# Limited Evidence for the Benefits of Exercise in Older Adults with Hematological Malignancies: A Systematic Review and Meta-Analysis

**DOI:** 10.3390/cancers16172962

**Published:** 2024-08-25

**Authors:** Mary Jarden, Sofie Tscherning Lindholm, Gudrun Kaldan, Charlotte Grønset, Rikke Faebo Larsen, Anders Thyge Steen Larsen, Mette Schaufuss Engedal, Marta Kramer Mikkelsen, Dorte Nielsen, Anders Vinther, Niels Abildgaard, Anders Tolver, Carsten Bogh Juhl

**Affiliations:** 1Department of Hematology, Center for Cancer and Organ Diseases, Copenhagen University Hospital, Rigshospitalet, Blegdamsvej 9, DK-2100 Copenhagen, Denmark; mette.schaufuss.engedal@regionh.dk; 2Health Research Unit, Center for Cancer and Organ Diseases, Copenhagen University Hospital, Rigshospitalet, Blegdamsvej 9, DK-2100 Copenhagen, Denmark; sofie.tscherning.lindholm.01@regionh.dk (S.T.L.); gudrun.kaldan.01@regionh.dk (G.K.); 3Department of Clinical Medicine, University of Copenhagen, Blegdamsvej 3B, DK-2200 Copenhagen, Denmark; dorte.nielsen.01@regionh.dk; 4Department of Physical and Occupational Therapy, Copenhagen University Hospital Hvidovre, Kettegård Allé 30, DK-2650 Hvidovre, Denmark; 5Department of Orthopedic Surgery, Copenhagen University Hospital Hvidovre, Kettegård Allé 30, DK-2650 Hvidovre, Denmark; 6Department of Occupational Therapy and Physiotherapy, Centre of Head and Orthopaedics, Copenhagen University Hospital, Rigshospitalet, Blegdamsvej 9, DK-2100 Copenhagen, Denmark; charlotte.groenset@regionh.dk; 7Research Unit of Hematology, Odense University Hospital, J. B. Winsløws Vej 4, DK-5000 Odense, Denmark; rikke.larsen.05@regionh.dk (R.F.L.); niels.abildgaard@rsyd.dk (N.A.); 8Center for Health Research, Copenhagen University Hospital, Rigshospitalet, Ryesgade 27, DK-2200 Copenhagen N, Denmark; anders.larsen@regionh.dk; 9Department of Oncology, Copenhagen University Hospital, Herlev and Gentofte, Borgmester Ib Juuls Vej 13, DK-2730 Herlev, Denmark; marta.kramer.mikkelsen@regionh.dk; 10QD Research Unit, Department of Occupational Therapy and Physiotherapy, Copenhagen University Hospital, Herlev and Gentofte, Borgmester Ib Juuls Vej 1, DK-2730 Herlev, Denmark; anders.vinther@regionh.dk (A.V.); carsten.bogh.juhl@regionh.dk (C.B.J.); 11Danish Cancer Institute, Strandboulevarden 49, DK-2100 Copenhagen, Denmark; atolve@cancer.dk; 12Department of Sports Science and Clinical Biomechanics, University of Southern Denmark, Campusvej 55, DK-5230 Odense, Denmark

**Keywords:** hematological malignancy, older patients, exercise, physical function, aerobic capacity, strength, health-related quality of life, symptoms

## Abstract

**Simple Summary:**

It is unknown whether different types of exercise effect physical function, quality of life, mental wellbeing, and symptoms in older patients with blood cancers. No studies specifically focusing on patients over 65 years were identified. However, across adult age groups, we found that exercise has small to moderate positive effects on physical function, aerobic capacity, muscle strength, quality of life, fatigue, pain, anxiety, and depression. These benefits were generally consistent regardless of age, except for physical function and pain, which favored younger adults. Overall, exercise improves physical function and quality of life and reduces symptoms in adults with blood cancers undergoing treatment, but the role of age remains uncertain.

**Abstract:**

Older patients receiving antineoplastic treatment face challenges such as frailty and reduced physical capacity and function. This systematic review and meta-analysis aimed to evaluate the effects of exercise interventions on physical function outcomes, health-related quality of life (QoL), and symptom burden in older patients above 65 years with hematological malignancies undergoing antineoplastic treatment. This review adheres to Cochrane guidelines, with the literature searches last updated on 27 March 2024, including studies with patients above 18 years. Screening of identified studies, data extraction, risk of bias, and GRADE assessments were performed independently by two authors. Meta-analyses evaluated the impact of exercise, considering advancing age. Forty-nine studies contributed data to the meta-analyses. Five studies included patients with a mean age above 60 years, and none included only patients above 60. Exercise interventions had moderate to small positive effects on QoL global (SMD 0.34, 95% CI [0.04–0.64]) and physical function (SMD 0.29, 95% CI [0.12–0.45]). Age did not explain the variability in exercise effects, except for physical function (slope 0.0401, 95% CI [0.0118–0.0683]) and pain (slope 0.0472, 95% CI [0.01–0.09]), which favored younger patients. Exercise interventions improve physical function and QoL and reduce symptoms in adults with hematological malignancies undergoing antineoplastic treatment; however, the influence of age remains inconclusive.

## 1. Introduction

The global increase in the aging population, in tandem with the delayed onset of most hematological malignancies, has led to a higher incidence of these cancers among older individuals [1,2]. Approximately half of all hematological malignancies occur in adults aged 65 years and older, and 70% of cancer-related deaths occur within this age group [3]. Overall survival in adults above 75 years old with hematological cancers, particularly acute leukemia, is low [4]. Older adults with hematological malignancies undergoing medical antineoplastic treatment experience challenges, including frailty and decline in physical and functional capacity, which can further burden their cancer treatment and overall recovery [5]. Among older patients with solid and hematological cancers, frailty is a prevalent concern associated with suboptimal therapeutic responses, increased treatment-related toxicity, and overall poorer survival outcomes [6]. Moreover, challenges like pain, fatigue, insomnia, and mood disturbances are prevalent among older cancer patients, significantly impacting their quality of life [7]. The estimated prevalence of cancer-related fatigue (CRF) is as high as 70% in individuals above 65 years [8]. This not only hinders everyday functioning but may also lead to significant disability and increased reliance on assistance for activities necessary for independent living [9].

Effective management of hematological malignancies in older patients requires a comprehensive, multidisciplinary approach that addresses both the disease and overall health and wellbeing [5,10]. In addition to the benefits provided by onco-geriatric interventions, which include screening for comorbidities and frailty to improve treatment outcomes, there is an increased interest in exploring nonpharmacological clinical interventions to improve health-related quality of life (QoL) and outcomes in older adults with solid and hematological cancers [11,12,13,14,15]. In recent years, growing recognition of the feasibility, safety, and potential benefits of exercise in improving clinical outcomes and QoL in patients with hematological malignancies [16,17] aligns with studies showing positive effects on physical capacity and function, psychological wellbeing, and the mitigation of fatigue and other treatment-related side effects in this patient population [18,19,20,21]. To guide exercise practices in older adults, several recommendations are available for clinical and research practices, such as the International exercise recommendations for older adults from 2021 [22], the American College of Sports Medicine (ACSM) exercise guidelines for older adults from 2009 [23], recommendations from 2015 for designing exercise clinical trials for older adults with cancer [24], and the 2019 roundtable on exercise guidelines for cancer survivors, which addresses issues of older cancer patients [25]. There remains, however, limited evidence supporting the benefits of exercise in older adults with hematological malignancies in these guidelines.

Consequently, there is a clear imperative to explore the impact of exercise in older adults with hematological cancers [26]. Given age-related physiological changes coupled with the complexities of hematological diseases and intensive treatments, there is a need for a comprehensive investigation into the potential benefits of exercise for this population. A systematic review and meta-analysis can provide insights into the effectiveness of exercise interventions on diverse outcomes and guide clinical strategies to optimize the health and wellbeing of older patients undergoing neoplastic treatment for hematological malignancies.

The aim of this systematic review and meta-analysis was to investigate the effect of exercise interventions on physical function, QoL, and symptom burden in older adults with hematological malignancies receiving medical antineoplastic treatments.

## 2. Materials and Methods

This systematic review and meta-analysis adheres to Cochrane guidelines [27] and is registered in The International Prospective Register of Systematic Reviews (PROSPERO) database (registration no. CRD42019130979 amended 24 May 2021 and 8 August 2023) and reported in accordance with the Preferred Reporting Items for Systematic Reviews and Meta-Analysis (PRISMA) guidelines [28,29].

### 2.1. Information Sources and Search Strategy

A systematic search was conducted in March 2019 across four databases: MEDLINE, EMBASE, Cochrane Central Register of Controlled Trials (CENTRAL), and CINAHL. No randomized controlled trials exclusively including participants above 65 years with hematological malignancies were identified (Appendix A). In October 2020, the search was expanded to include individuals above 18 years old, and an additional database, Science Citation Index Expanded/Social Sciences Citation Index (SCIE/SSCI), was added (Appendix A). On 27 March 2024, an updated search was carried out in MEDLINE, EMBASE, CENTRAL, CINAHL, and SCIE/SSCI (Appendix A). Using the PICO framework (Problems, Interventions, Comparisons and Outcomes) [30], the searches included three main areas: (1) exercise-based interventions, (2) hematological malignancies during medical antineoplastic treatment, and (3) randomized controlled trials. The searches involved a combination of MeSH/Thesaurus/indexed terms and relevant synonyms (Appendix A).

### 2.2. Eligibility Criteria

Studies included were randomized controlled trials, randomized pilot trials, randomized crossover trials, and randomized feasibility studies with adult participants (≥18 years) diagnosed with hematologic malignancies undergoing medical antineoplastic treatment, such as chemotherapy, chemoradiation, stem cell transplantation including allogeneic hematopoietic stem cell transplantation (allo HSCT), autologous stem cell transplantation (autologous HSCT), and high-dose chemotherapy with stem cell support. Studies that included participants just prior to starting treatment and up to six months post-treatment were considered. All types of exercise-based interventions and complementary alternative medicine (CAM) such as yoga, compared to standard care were considered for inclusion, including multimodal interventions where the exercise component constituted at least 50% of the intervention sessions. Interventions could be initiated prior to, during hospitalization or outpatient clinic visits, or after discharge, including home-based training. All durations of interventions were included. For comparator(s) or control groups, all types, whether active or inactive, were considered. Studies included measured outcome effects. No restrictions were applied for language or publication date.

### 2.3. Data Collection Process

Records were imported to Covidence, a data management program (www.covidence.org; https://app.covidence.org/reviews/59783, accessed on 14 June 2019; https://app.covidence.org/reviews/122435, accessed on 11 November 2020; https://app.covidence.org/reviews/425529; accessed on 27 March 2024) for duplicate removal, screening, and data extraction. Two authors independently performed screening of identified studies and data extraction on author details (year, country), population characteristics (age, gender, diagnosis, treatment), study characteristics (design, sample size, recruitment, timing, retention, feasibility, adherence, safety), exclusion criteria, intervention characteristics (delivery mode, type, length, intensity), control group specifics, and outcome measurements. Outcome data were extracted at baseline and post-intervention, the time point nearest to the intervention completion date.

To identify ongoing exercise-based randomized trials, a comprehensive search was carried out in ClinicalTrials.gov, accessed on 13 November 2023, including only patients above 60 years with hematological malignancies undergoing medical antineoplastic treatment (Appendix A). Information extracted includes trial identifier, author information, population characteristics (age, gender, diagnosis, treatment modalities), study design, type and duration of exercise intervention, primary outcomes, and current study status.

### 2.4. Outcomes

The primary outcomes were changes in physical function (e.g., 6-min walk test (6MWT)) and QoL global (e.g., The European Organization for Research and Treatment of Cancer Core Quality of Life Questionnaire (EORTC QLQ-C30)). Measurements of QoL global included both generic and disease-specific patient-reported outcome measures (PROMs).

Secondary outcomes included changes in aerobic capacity (e.g., Volume Oxygen Peak, (VO_2_ Peak)), muscle strength (e.g., Sit to Stand test (STS)), body composition (e.g., fat-free mass measured by Dual X-ray Absorptiometry (DEXA)), and PROMs evaluating physical activity (e.g., International Physical Activity Questionnaire (IPAQ)), physical, emotional, and functional well-being (e.g., EORTC QLQ-C30). Patient-reported symptoms comprised anxiety (e.g., Hospital Anxiety and Depression Scale (HADS)), depression (e.g., HADS), fatigue (e.g., The Functional Assessment of Chronic Illness Therapy Fatigue Scale (FACIT-F)), and pain (e.g., EORTC QLQ-C30), in addition to an assessment of feasibility and safety.

For the meta-analysis, and for each outcome, distinct measurement instruments and tests were chosen, prioritizing those most widely applied for assessing the outcomes of exercise interventions, considering validity and reliability (Appendix A). For each study, a single measurement instrument was selected for each outcome.

### 2.5. Risk of Bias Assessment

Risk of bias for each study was independently assessed by two of six authors (STL, GK, RFL, CG, MSE, MJ) using the Revised Cochrane risk-of-bias tool for randomized trials (RoB 2) [31]. Any disagreements were resolved through discussions with MJ.

### 2.6. Certainty of Evidence

The certainty of evidence for each outcome was assessed using the Grading of Recommendation, Assessment, Development and Evaluation tool (GRADE) [32] by three authors (STL, GK, MJ) utilizing the GRADEpro Guideline Development Tool [Software] (McMaster University and Evidence Prime, 2024; Available from gradepro.org; accessed on 21 August 2024). The certainty of evidence was evaluated in five domains: study quality, inconsistency, indirectness, imprecision, and publication bias. Risk of publication bias (small study bias) was assessed using Egger’s test and illustrated with a funnel plot for all outcomes, which was inspected for asymmetry.

### 2.7. Data Synthesis and Analysis

The effect size of each intervention was estimated as the standardized mean difference (SMD) with a 95% CI using the restricted maximum likelihood model (REML). The SMD was calculated as the difference in mean change between the intervention and control group, divided by the pooled standard deviation (SD), and adjusted to Hedges’ g to account for a small overestimation of the effect in small studies. A random effects model was used for all meta-analyses, as it was expected that participants, interventions, and outcome measures might differ between studies [27]. To facilitate a unified analysis across multiple outcomes, we analyzed changes in measurement scores from baseline to post-test, and studies not reporting change scores were converted to change scores by using a correlation of 0.6. Interpretation of the SMD adhered to Cohen’s recommendations: small effect was defined as 0.2, moderate effect as 0.5, and large effect as 0.8 [27]. 

Heterogeneity was evaluated using the I^2^ index, and the between-study variance as tau-square (t^2^). Meta-regression analyses employing STATA version 18 (StataCorp 2023. Stata Statistical Software: Release 18. College Station, TX, USA) were conducted to assess potential influences of mean age of participants on effects in the meta-analysis. The number of participants for each outcome corresponds to those who completed the study with data, which may not necessarily be the same as the total number included in the studies.

In subgroup analyses, we examined covariates including mean age as a dichotomous variable (<60 years or >60 years), diagnosis (acute leukemia, lymphoma, multiple myeloma, mixed diagnoses), and antineoplastic treatment (chemotherapy, allogeneic HSCT, autologous HSCT or both, different treatments). Given the anticipated heterogeneity in exercise interventions, we categorized the exercise interventions based on type (aerobic, strength, combined aerobic and strength, either aerobic or strength exercise, CAM), timing of exercise (before, during, after treatment, or combination), delivery mode (supervised, partly supervised, unsupervised), individual or group-based, and extensiveness of intervention, and risk of bias (high risk, some concerns, and low risk). Extensiveness of exercise interventions was categorized based on number of sessions, session length, and intensity, inspired by guidelines from the American College of Sports Medicine [25,33] and two systematic reviews of exercise interventions [34,35]. Each exercise intervention was categorized as less, moderate, or extensive. The categorization process was collaboratively conducted by three authors (MJ, STL, GK), with STL providing expertise as a physiotherapist.

## 3. Results

The searches identified 5550 records (Appendix A). After removing duplicates and screening, 49 studies were included [36,37,38,39,40,41,42,43,44,45,46,47,48,49,50,51,52,53,54,55,56,57,58,59,60,61,62,63,64,65,66,67,68,69,70,71,72,73,74,75,76,77,78,79,80,81,82,83,84], reported in 51 publications, with two of these being follow-up analyses [85,86]. All studies were included in the narrative review (Figure 1).

### 3.1. Study Characteristics

In total, 3494 patients were included, with sample sizes ranging from 17–711 (Table 1). In 41 studies, the mean age was 50 years (range 18–90) [36,37,38,39,40,41,42,43,44,45,47,48,49,50,51,54,55,56,57,58,59,60,62,63,64,65,66,68,69,70,71,72,73,74,75,77,78,79,80,82,84], and in seven studies, the median age was 53 years [46,52,61,67,76,81,83]. Only five studies reported a mean age of participants ≥ 60 years (n = 359) [47,57,58,67,68]. Females constituted 43% of participants.

Studies included mixed hematological cancer diagnoses (n = 30) [36,39,40,41,42,46,52,53,54,56,59,60,61,62,64,65,66,69,70,71,72,73,74,75,76,77,78,79,82,83], acute leukemia (n = 9) [37,38,43,44,45,55,63,80,81], multiple myeloma (n = 5) [49,50,57,67,68], and lymphoma (n = 5) [47,48,51,58,84]. Studies comprised patients undergoing chemotherapy (n = 20) [36,38,41,43,44,45,47,48,51,55,58,59,63,66,70,71,75,78,81,84], allo-HSCT (n = 10) [40,52,53,62,64,69,72,77,82,83], autologous HSCT (n = 4) [57,60,68,73], allogeneic and autologous HSCT (n = 6) [42,56,61,65,74,76], and different treatments (n = 8) [37,39,46,49,50,54,67,79]. One study did not specify the type of medical antineoplastic treatment [80].

Thirty-four studies were published within the last decade (2013–2024), most originating from the USA (n = 14) [43,46,48,49,50,52,53,56,57,59,61,74,77,83] and Germany (n = 9) [39,40,70,71,72,76,78,81,82]. The 49 trials were mainly designed as 2-arm (n = 46) [36,37,38,39,40,42,43,44,45,46,47,48,49,50,51,52,53,54,56,57,58,59,60,62,63,64,65,66,67,68,69,70,71,72,73,74,75,76,77,78,79,80,82,83,84], with 29% being pilot (n = 14) [39,43,45,46,49,55,57,59,60,68,70,71,81,83] or feasibility (n = 3) [37,53,76] randomized trials.

The control groups received usual care in 28 studies [38,39,40,43,45,47,48,49,50,51,52,57,59,60,62,63,64,65,67,68,70,73,74,77,78,79,80,84]. To counteract socio-psychological bias, 20 studies [36,38,41,42,44,46,53,54,55,56,58,61,66,71,72,75,76,81,82,83] implemented physical or attention activities (activity tracker, diary, phone calls) in the control groups.

### 3.2. Exercise Interventions

The type, intensity, length, and duration of the exercise interventions are presented in Table 1. The most common type of exercise intervention was a combination of aerobic (cycling, running, walking) and resistance (machines, resistance bands, body weight) exercise (n = 27) [36,37,38,39,40,41,42,43,49,50,54,57,58,60,62,63,65,66,67,68,69,70,73,76,77,78,82,86], followed by aerobic exercise (n = 11) [44,46,51,52,55,61,74,79,80,81,83], resistance exercise (n = 6) [55,56,71,72,75,81], and CAM practices such as yoga, Qi gong, relaxation, and breathing exercises (n = 7) [45,47,48,53,59,64,84]. The duration of exercise interventions in the included studies varied (range 1–36 weeks), with 12 weeks being the most common (n = 11) [37,51,53,54,58,59,60,63,65,79,80]. The most common frequency of intervention was three times a week (n = 11) [51,54,58,63,67,68,71,74,77,80,81], and some studies allowed variation and range in frequency, i.e., 2–3 times a week or 3–5 times a week (n = 5) [49,50,53,55,59]. The timing of the intervention was mostly during treatment (n = 21) [38,39,40,41,43,44,45,47,49,50,55,62,63,66,70,71,72,75,78,81,84], and after treatment (n = 13) [37,42,46,53,54,57,60,64,65,67,73,77,80].

The exercise interventions were mainly supervised (n = 27) [38,39,40,41,42,43,44,48,51,55,62,63,64,65,66,69,70,71,72,73,74,75,76,77,78,80,81]. Twenty-four took place at the hospital [38,39,40,42,43,44,45,48,51,62,63,64,66,69,70,71,72,74,75,76,77,78,80,81], and 12 were home-based [46,47,49,50,53,58,59,60,61,79,83,84]. Most studies offered individualized interventions (n = 44) [36,37,38,39,40,41,43,44,45,46,47,49,50,52,53,54,55,56,57,58,59,60,61,62,64,65,66,67,69,70,71,72,73,74,75,76,77,78,79,80,81,82,83,84,86]. The level of extensiveness of exercise interventions was predominantly moderate (n = 32) [37,38,39,40,43,49,50,51,52,54,55,56,57,58,60,61,62,65,67,69,70,71,72,74,75,76,77,79,80,81,82,83].

### 3.3. Effects of Exercise Interventions on Primary Outcomes

Data on physical function and QoL global were available for meta-analysis in 25 and 29 studies, including 1219 and 1447 participants, respectively. Exercise had significant moderate to small effects on physical function, SMD 0.29 (95% CI 0.12–0.45); I^2^; 48,17% and QoL global, SMD 0.34 (95% CI 0.04–0.64); I^2^: 87.39% (Table 2 and Appendix A).

Meta-regression analyses on exercise benefits on age for QoL global showed a non-significant coefficient 0.0247 (95% CI −0.0140–0.0634, *p* = 0.210), favoring older patients, and physical function showed a significant negative coefficient 0.0401 (95% CI 0.0118–0.0683, *p* = 0.005), favoring younger patients (Appendix A).

Subgroup analyses on QoL global showed the largest effect in the age group over 60 years (SMD 1.29), and for physical function, it was below 60 years (SMD 0.32). Results for subgroup analyses for physical function and QoL global are presented in Appendix A, respectively.

### 3.4. Effects of Exercise Interventions on Secondary Outcomes

The largest effects of exercise were observed in aerobic capacity, SMD 0.53 (95% CI 0.27–0.79) and in muscle strength (SMD 0.47 (95% CI 0.17–0.78)). Moderate to small significant symptom-related benefits were found in fatigue (SMD 0.44), pain (SMD 0.43), and depression (SMD 0.37), and a small non-significant effect on anxiety (SMD 0.21). Significant moderate to small effects on QoL emotional (SMD 0.33), QoL functional (SMD 0.33), and QoL physical domains (SMD 0.32), as well as non-significant on the physical activity outcome (SMD 0.32), were observed. There was a small significant effect in body composition (SMD 0.20), all favoring exercise interventions (Table 2 and Appendix A).

Except for pain (coefficient 0.0472 (95% CI 0.0078–0.0866, *p* = 0.019), favoring younger patients, age did not influence secondary outcomes (Appendix A).

Subgroup analyses showed that patients aged above 60 years yielded the largest effect on anxiety (SMD 0.44), fatigue (SMD 0.97), muscle strength (SMD 0.98), pain (SMD 1.31), QoL emotional (SMD 0.95), and QoL physical (SMD 1.50). Results for subgroup analyses are presented in Appendix A.

### 3.5. Feasibility, Adverse Events, Adherence and Exclusion Criteria

Feasibility, adverse events, and adherence are presented in Table 3, and exclusion criteria in Table 1. One third of the studies (n = 16) did not provide information on adverse events (AE) [37,39,45,46,48,50,52,56,57,64,69,74,77,80,82,83]. Twenty-six studies reported no AE [36,40,41,42,43,44,47,49,53,55,58,59,60,61,62,65,66,67,70,71,75,76,78,79,81,84], and seven studies reported non-serious AE such as back, hip, and knee pain, and cardiorespiratory and gastrointestinal symptoms [38,51,54,63,68,72,73]. In the five studies with mean/median above 60 years, exercise interventions were tolerated, with three studies reporting no AE [47,58,67], one reporting non-serious AE [68], and one not reporting AE [57] (Table 3).

Thirty-three studies estimated sample sizes either through non-formal estimation or power calculations [36,37,38,39,40,41,42,43,45,46,47,48,50,51,55,58,61,62,63,64,65,66,67,70,73,75,77,78,79,80,81,83,84]. Twenty-two studies met sample size estimation [36,38,39,40,41,45,46,47,48,51,58,61,62,63,64,65,66,75,79,80,81,84], and eleven were underpowered [37,42,43,50,55,67,70,73,77,78,83]. Seventeen studies did not report sample size estimation [44,49,52,53,54,56,57,59,60,68,69,71,72,74,76,82]. Of those reporting (n = 41), a total of 7262 patients were screened for eligibility across the studies. Of these, 3552 were included in the studies, and 2924 participants completed the post-test, resulting in a retention rate of 82.3%.

Adherence to exercise was reported in 28 studies with a mean exercise adherence of 70% (range 15–100%), though 21 studies did not report adherence [39,40,42,44,49,50,53,55,57,58,60,61,62,64,68,69,70,75,76,80,83]. Two studies with patients mean age above 60 reported high adherences to the exercise intervention (range 75–96%) [47,67].

Inclusion and exclusion criteria varied across the 49 studies. The number of individual exclusion criteria per study was mean five (range 1–17), with the most frequent exclusion criteria being cardiovascular disease (n = 24), malignancy (n = 23), musculoskeletal disorder (n = 22), and mental health challenges (n = 18). Seven studies did not report exclusion criteria [42,44,49,57,69,73,82]. Despite detailed descriptions of inclusion and exclusion criteria in most studies, several did not report reasons for study exclusion (n = 14) [39,40,42,45,46,48,49,50,61,64,66,72,77,80]. In studies that reported (n = 34), the three most prevalent reasons for study exclusion were related to a) medical and health status: comorbidity, unstable conditions, adverse effects, malignancy, symptoms, low blood counts; b) physical performance: mobility challenges and frailty; and c) mental health issues: psychological instability and cognitive impairment. Some studies were not transparent, detailed, or systematic in reporting reasons for exclusion; and in some cases, it was just stated that participants did not meet inclusion criteria without providing explanation.

### 3.6. Risk of Bias in Individual Studies and across Studies

Risk of bias assessment for each study is reported in Figure 2. Most (n = 23) were assessed to have some concerns [36,42,43,44,45,47,54,58,60,61,62,63,65,66,68,71,73,77,79,81,82]. Eighteen were evaluated as high risk [37,46,48,49,50,52,53,55,56,59,64,69,70,72,74,76,78,86] and ten as low risk [38,39,40,41,51,57,67,75,80,83]. High risk and some concerns were mainly due to Domain 2, deviations from the intended intervention, or Domain 3, missing outcome data.

### 3.7. Quality of Evidence (GRADE)

Low to very low quality of evidence was found for a small effect on primary outcomes; physical function and QoL global, as evidence was downgraded due to high risk of bias, inconsistency, and risk of publication bias (Table 2). For secondary outcomes, only body composition showed high quality of evidence but for a small effect, while the remaining secondary outcomes ranged from very low to moderate levels of evidence.

### 3.8. Ongoing Studies Registered in Clinical Trials

There are currently four ongoing randomized exercise trials for older adults with hematological malignancies. Two studies recruit patients over 60 years, and the other two recruit patients over 65 and 70 (Table 4).

## 4. Discussion

This systematic review and meta-analysis is a synthesis of data on the effect of exercise interventions in adults with hematological malignancies undergoing medical antineoplastic treatments. Notably, we did not find any studies that only included adults aged 65 years and older. Therefore, the scope was expanded to include all adult age groups (above 18 years), and we found exercise interventions to have significant positive benefits on outcomes, irrespective of age. Among the 49 studies analyzed, only five included a sample with mean or median age over 60 years, indicating a scarcity of evidence specifically addressing the older population with hematological malignancies. Likewise, Knowles et al. 2022 in a review of reviews of older people with cancer did not find any systematic reviews including a population over 65 years [87]. Similarly, Mikkelsen et al. 2020 in a systematic review of exercise interventions in older cancer patients identified only four studies that included a sample of patients above 65 years and found inconclusive evidence regarding benefits of exercise in this age group [88].

Given the rising prevalence of hematological malignancies among older adults and the increasing demand for more comprehensive approaches to treatment and care for this population [5,10], evidence regarding exercise remains notably lacking for older adults. We identified four ongoing exercise-based randomized trials in patients with hematological malignancies above 60 years registered in clinicaltrials.gov, accessed on 13 November 2023 (Table 4. These ongoing studies, though few, highlight the growing need to understand the role of exercise in improving physical function and QoL, and mitigating symptoms in older adults undergoing antineoplastic treatment.

In this review, we found exercise to have small to moderate certainty evidence across most outcomes across age groups. Likewise, several systematic reviews and meta-analyses, each with relatively few studies, also found exercise benefits in adults with hematological malignancies. Abo et al. (2021) reported moderate evidence in functional capacity, QoL global, and fatigue [20]. Moore et al. (2023) found evidence for improvements in physical function [21]. Yang et al. 2022 found significant improvements in QoL, emotional functioning, and pain [16]. Knips et al. (2019) found low-certainty evidence on depression and anxiety and moderate evidence for fatigue [18]. Research across hematological diagnoses and treatments consistently indicates that exercise positively impacts a broad range of outcomes in adults generally. However, our meta-regression analysis found that impact on QoL global favored advancing age, while physical function and pain favored younger adults. Our subanalyses, which should be interpreted with caution due to the limited number of studies included, showed that those aged above 60 had the largest effects on QoL domains (global, emotional, and physical), muscle strength, pain, fatigue, and anxiety, whereas those below 60 years showed the largest effects in physical function, aerobic capacity, QoL functional, and body composition.

Exercise interventions in our systematic review were reported as feasible and safe in 24 studies, with seven studies documenting nonserious AEs. However, one-third of the studies (n = 16) did not report on AEs. Similarly, GroBek et al.’s (2023) systematic review on the safety and feasibility of exercise interventions in patients with hematological cancer (12 studies), which includes the same studies as our systematic review, concluded that exercise is feasible and safe in the studies reporting on it, but also found a similar lack of AE reporting, and when provided, the information was often insufficient [17]. We recommend systematic monitoring and adequate documentation of feasibility and safety, including AEs, especially when providing exercise to hematological patients, given the potential comorbidities. This approach would facilitate targeted delivery of exercise interventions, not only to older adults but also to specific patient populations experiencing comorbidities.

Studies included in our review did not exclude patients based on older age; however, most studies used stringent exclusion criteria that targeted comorbidities and reduced function and performance inherent to this age group. This is evident in the detailed and high number of inclusion and exclusion criteria (up to 16), but importantly, reasons for exclusion were either not reported at all or reported with lack of detail, a pattern consistent with other exercise reviews of older cancer patients [87,88]. The rigorous exclusion criteria may contribute to the underrepresentation of older patients with comorbidities in exercise studies, as pointed out in several reviews [89,90,91]. However, there are exceptions, as in a feasibility pilot strength training program among an older population of patients (mean 68 years) with multiple myeloma, which included those with osteolytic lesions, establishing both feasibility and safety [92]. Older adults with the greatest need for exercise may not be considered suitable to participate due to strict safety criteria. This emphasizes the challenge of striking a balance between prioritizing safety and unintentionally excluding patients who could derive benefits from exercise.

Future exercise studies are likely to be designed to target the individual needs of older cancer patients, ultimately incorporating a more representative sample of patients. This will increase the generalizability of findings to the broader population of patients with hematological malignancies. Rosko et al. (2022) tested the feasibility of implementing an exercise intervention in older adults with hematologic malignancy and found a higher completion rate with in-person, physiotherapist supervision compared to at-home independent exercise. Though these older patients were motivated to follow a structured exercise program, health status changes were the main barrier to exercise [93].

In our review, adherence to exercise was predominantly high (72%). Two studies with a mean sample over 60 years reported high adherence (range 75–96%) [47,67]. Slightly lower, Mazzoni et al. (2020) found exercise adherence in a mixed cancer diagnosis group was ≥50% [94]. In our review, 20 studies did not provide adherence rates. Adherence is addressed in an umbrella scoping review by Collado-Mateo et al. (2021) of 53 studies of chronic disease and older individuals and identified multiple modifiable factors influencing adherence to exercise in this population [95]. Emphasizing factors that can improve adherence when planning and designing exercise interventions for the older population may improve feasibility and safety, and ultimately lead to better outcomes.

The extensiveness of exercise interventions varied across studies in this review, with moderate extensiveness (n = 32) being the most frequently designed intervention for the population with hematological malignancy. This pattern remained consistent in five studies where the mean or median age was above 60 years (n = 3). Conducting extensiveness rating across studies used in systematic reviews by Andersen et al. (2022) [34] and Ramírez-Vélez et al. (2021) [35] provides a more uniformed overview of the type of intervention that may be most beneficial for different populations. It is important to emphasize the significance of tailored exercise interventions for older adults with hematological malignancies, considering a more comprehensive supportive approach that takes individual needs and preferences into account, including not only physical and QoL outcomes but also clinical outcomes.

Although exercise recommendations for adults with cancer are available, evidence specifically for older cancer patients is limited and based on only a few studies [25]. There are no evidence-based recommendations specifically tailored to older patients with hematological cancers. However, it is suggested that greater focus on exercise in the older cancer population to optimize exercise delivery, participation, safety, and efficacy is justified [26]. Recognizing and understanding the benefits of exercise for older patients with hematological malignancies can be an important initiative to integrate into clinical practice.

Further research is needed on exercise interventions specifically designed for older patients with hematological cancers, including those with typical age-related comorbidities. Emphasis should be on addressing their individual needs, considering exercise types and contexts, and implementing strategies to improve recruitment accrual, minimize attrition, and enhance adherence rates. Involvement of patients in the design of these studies is important for relevance and overall quality of the study. Colton et al. (2022) captured the voices of older patients with hematological cancer, highlighting their specific needs and preferences on diet and exercise behavior [96]. As per our review, Kilari et al. (2016) also emphasized the notable lack of exercise studies specifically tailored to the older cancer population. To effectively inform future practices for this population, merely extracting results and insights from studies of a wide age range is not adequate for developing exercise prescriptions for older patients with hematological cancer. Future studies should incorporate outcomes beyond physical function and PROMs such as QoL and symptom burden and should also provide knowledge on the clinical benefits of exercise including admission days, complications, and survival [24].

### Strengths and Limitations

This systematic review and meta-analysis provides the largest overview of exercise studies in a total of 3494 patients with hematological cancer undergoing neoplastic treatments. Notably, no study exclusively included participants over 65 years, or even 60 years. Consequently, drawing definitive conclusions regarding the benefit of exercise for older patients based solely on our meta-regression analysis and sub-analyses is not possible. Despite the inclusion of a large number of studies in this meta-analysis, we need to consider the substantial heterogeneity across studies, regarding variations in diagnosis groups, types of antineoplastic treatments, and the extensiveness of exercise interventions. The secondary outcomes and subgroup analyses are exploratory; therefore, interpreting results at a significance level of 0.05 can potentially pose a risk of false positive results (mass significance). We conducted three systematic searches but also acknowledge the possibility of having overlooked relevant studies in these repeated searches.

## 5. Conclusions

Exercise in adults with hematological malignancy undergoing neoplastic treatment provides significant benefits in physical function, aerobic capacity, muscle strength, physical activity, body composition, and QoL. Symptoms of depression, anxiety, fatigue, and pain were significantly reduced. The influence of age on the benefits of exercise remains inconclusive. The absence of studies specifically in older adults is staggering given that hematological malignancies predominantly affect the older population. Notably, there is a modest number of ongoing exercise-based randomized studies in older patients with hematological malignancy (clinicaltrials.gov, accessed on 13 November 2023), and this reflects the increasing interest and urgent need for evidence-based guidelines tailored to this population.

## Figures and Tables

**Figure 1 cancers-16-02962-f001:**
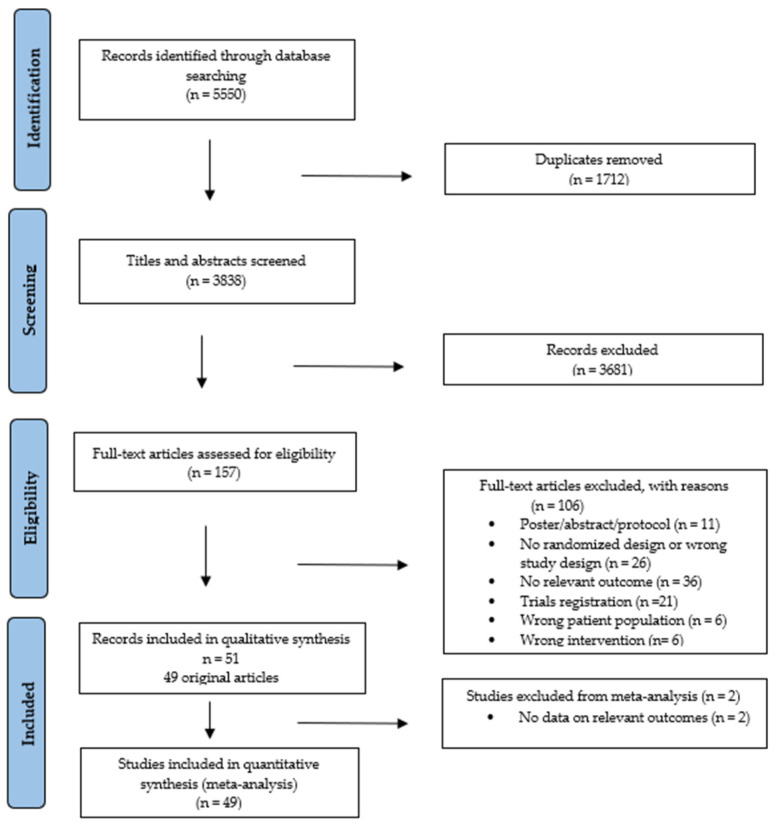
PRISMA flowchart of study selection process.

**Figure 2 cancers-16-02962-f002:**
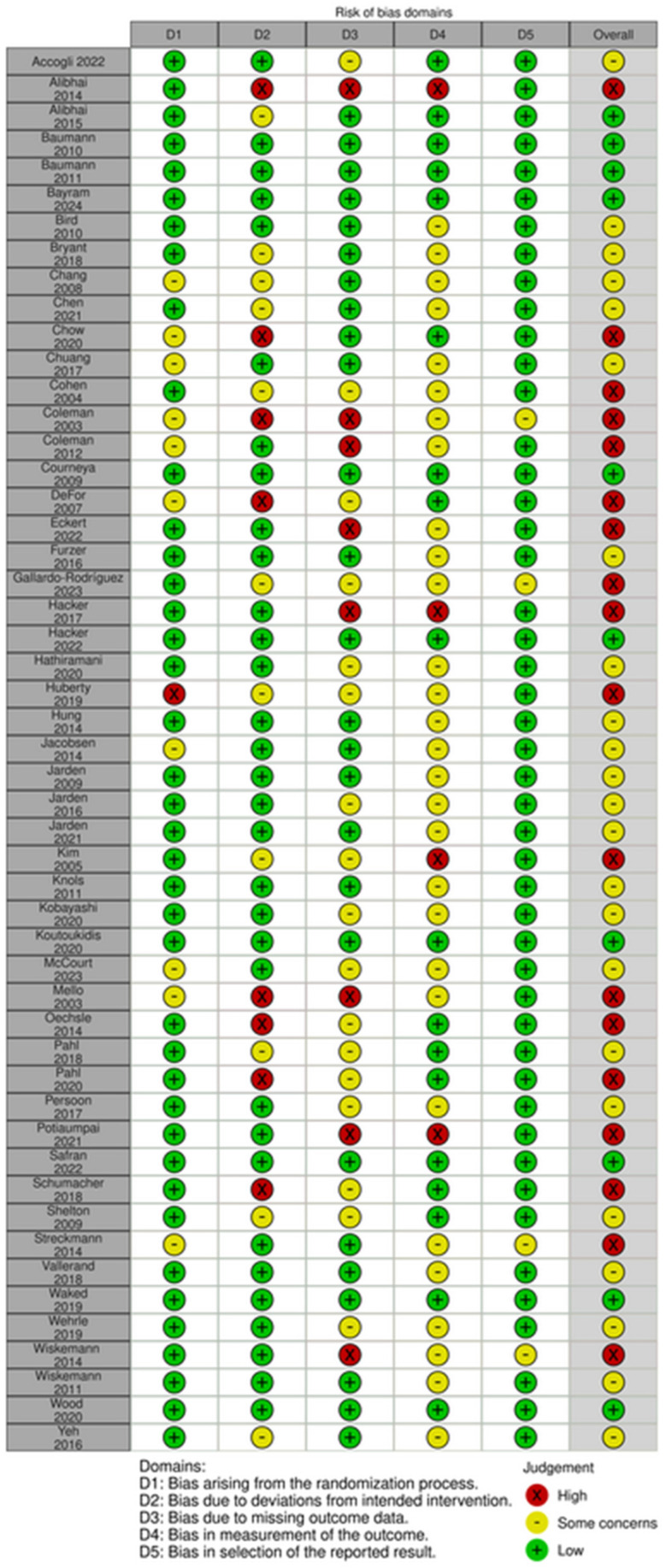
Risk of Bias [36,37,38,39,40,41,42,43,44,45,46,47,48,49,50,51,52,53,54,55,56,57,58,59,60,61,62,63,64,65,66,67,68,69,70,71,72,73,74,75,76,77,78,79,80,81,82,83,84,85,86].

**Table 1 cancers-16-02962-t001:** Characteristics of included studies.

Author, Year, Country	Diagnose	Study Design	Sample Size, n IG/CG Female (%)	Age Mean, (Range) Median, (Range)	Exclusion Criteria	Timing of Intervention	Intervention Group		Control Group
							Type: T Intensity: I Length: L Duration: D	Extensiveness	Type: T Intensity: I Length: L Duration: D
Accogli 2022 Italy [36]	Lymph, Leuk, MM	RCT	46 (23/23) (47.8)	Mean 59.9 Median IG: 66.7 (51.3–72.1) CG: 60.4 (49.9–67.5)	Poor prognosis (<12 months) and clinical conditions hindering participation (e.g., dementia, psychiatric pathology, blindness)	Before, during and after chemotherapy	**Partly supervised** Supervised, hospital-based: T: Therapeutic education I: Individual intensity L: 2 × 60 min (group) and face-to-face (individual) 6 × 20 min 1x/week or every 2 weeks Unsupervised, homebased: T: Individual physical exercise D: 8 weeks	Less	T: Educational therapeutic group sessions L: 2x in total
Alibhai 2014 Canada [37]	AML	RCT Feasibility	38 (21/17) (55.3)	Mean 56.1 IG: 53.9 CG: 58.8	Another active malignancy, life expectancy < 3 months, severe or unstable cardiorespiratory or musculoskeletal disease, awaiting HSCT and regular participation in a moderate-vigorous PA program	After HSCT or chemotherapy	**Partly supervised** Supervised, hospital-based: T: Workout and education (group) L: 1.5 h/week Unsupervised, home-based: T: Aerobic, resistance and flexibility components I: Moderate intensity L: 30 min, 3–5x/week D: 12 weeks	Moderate	T: UC, Usual level of PA
Alibhai 2015 Canada [38]	AML	RCT	81 (57/24) (45.7)	Mean 57 (23–80) IG: 58 CG: 52 Median 59 >60: IG, n = 32 CG, n = 7	Another active malignancy, life expectancy < 1-month, significant medical comorbidity that would preclude exercise, uncontrolled pain, hemo-dynamic instability	During chemotherapy	**Supervised**, hospital-based: T: Individualized aerobic (treadmill, hall walking, stationary cycling), resistance (body weight, bands, free weights) and flexibility training I: RPE of 3–6, equivalent to 50–75% of HRR L: 30–60 min, 4–5x/week	Moderate	T: UC, Suggestions to walk regularly and weekly document on tracking sheets
Baumann 2010 Germany [39]	AML, ALL, CML, MM, NHL/CLL, MDS/MPS, Solid tumor, Immuno-deficiency	RCT Pilot	64 32/32 (45.4)	Mean IG: 44.9 CG: 44.1	Severe orthopaedic illness of legs, severe heart failure (NYHA III-IV), metastatic bone disease, thrombocytopenia (≤30 × 10^9^/L) and/or acute somatic complaint (e.g., infection, fever, acute bleeding)	During ASCT, allo-HSCT or chemotherapy	**Supervised,** hospital-based: T: Aerobic (ergometer) and ADL training (walking, stepping, stretching) I: Aerobic: 80% of achieved watt load in WHO-test, ADL-training: Borg scale: “slighty strenuous”-“strenuous” L: Aerobic 10–20 min + ADL 20 min/2x daily, ADL 5x/week D: Mean: 26.6 days	Moderate	T: UC, Standard mobilization program
Baumann 2011 Germany [40]	AML ALL, CML, CLL, MPS, MDS, CMML, MM, PID	RCT	47 (24/23) (51.5)	Mean IG: 41.4 CG: 42.8	Severe cardiac disease (NYHA III-IV) or orthopaedic illness of the legs, bone metastases, thrombocytopenia (≤10 × 10^9^/L) or acute bleeding, respectively, and/or acute health or somatic complaints (e.g., infection, fever)	During allo-HSCT	**Supervised,** hospital-based: T: Aerobic (ergometer) and ADL training (walking, stepping, stretching) I: Aerobic: 80% of achieved watt load in WHO-test, ADL-training: Borg scale: “slighty strenuous”-”strenuous L: Aerobic: 10–20 min + ADL 20 min, 1–2x/day D: Mean: 56.1 days	Moderate	T: UC, Standard PT
Bayram 2024 Turkey [41]	ALL, AML, Biphenotypic Leuk, MDS, NHL, Burkitt Lymph, CNS Lymph, Myelofibrosis, Thalassemia major, MM	RCT	30 (15/15) (26.7)	Mean IG: 45.67 CG: 52.07	Orthopaedic, neurological, or cognitive disease affecting functional capacity, psychiatric disorders, pneumonia, acute infections, sepsis, and pulmonary diseases	During HSCT	**Supervised,** hospital-based: T: Aerobic (arm ergometer), resistance (free weights) and inspiratory muscle (inspiratory pressure device) exercises I: Aerobic: 50–80% of HR. Resistance: 4–6 on modified Borg scale, 3 sets of 10 reps. Inspiratory: 30% of max inspiratory pressure L: Aerobic: 10–30 min, 1x/day, 5 days/week. Resistance: 10–15 min, 5 days/week. Inspiratory: 15 min, 2x/day, 5 days/week D: During inpatient period. Mean: 25.2 days	Extensive	T: Aerobic and resistance exercises I: As IG L: As IG D: During inpatient period. Mean: 21.33 days
Bird 2010 UK [42]	Leuk, Lymph, MM, other	RCT	58 (29/29) (34.5)	Median 55 IG: 57 CG:52	NR	After ASCT or allo-HSCT	**Supervised,** hospital-based: T: Circuit training exercise, relaxation, and information (group) I: NR L: 1x/week D: 10 weeks	Less	T: UC, Self-managed program: information leaflets and home-based exercise program
Bryant 2018 USA [43]	AML, ALL	RCT Pilot	18 (9/9) (29.4)	Mean IG: 52 (34–67) CG: 49 (28–69) Median IG: 58 (34–67) CG: 48 (28–69)	Cardiovasc. disease, acute or chronic respiratory disease, acute or chronic bone, muscle, or joint abnormalities, altered mental state, dementia or any other psychological condition, another active malignancy, active bleeding, acute thrombosis, ischemia, hemodynamic instability, or uncontrolled pain	During chemotherapy	**Supervised,** hospital-based: T: Aerobic (walking or stationary bike) and resistance training (resistance band) I: Aerobic: 50–70% of HRR. Resistance: Increased from lighter to heavier resistance, 10 RM L: 20–40 min, 2x/day, 4x/week D: 4 weeks	Moderate	T: UC
Chang 2008 Taiwan [44]	AML	RCT	24 (12/12) (45.5)	Mean IG: 49.4 CG: 53.3	NR	During chemotherapy	**Supervised,** hospital-based: T: Walking exercise program I: A speed to reach target HR (resting heart rate plus 30) L: 12 min, 5x/week D: 3 weeks	Less	T: Nurse-led control L/D: 1x/day, 5 days/week, 3 weeks
Chen 2021 China [45]	AML, ALL	RCT Pilot	30 (15/15) (58.6)	Mean IG: 40.2 CG: 37.6	Medical conditions in arms, legs, or abdomen, paralysis, or disability and intended to receive HSCT in the next 3 months	During chemotherapy	**Partly supervised**, hospital-based T: Individualized self-help relaxation exercises I: NR L: 30 min, 2x daily D: 4 weeks	Less	T: UC
Chow 2020 USA [46]	Leuk, Lymph, other	RCT Pilot	41 (24/17) (48.8)	Median 45.1 (20.2–54.8) IG: 44.0 (20.9–54.0) CG: 46.0 (20.2–54.8)	Pre-existing ischemic heart disease or ongoing symptomatic cardiomyopathy, active cGvHD, pregnant	After ASCT, allo-HSCT	**Unsupervised**, home-based: T: Individualized, multiple mHealth app-based lifestyle counselling and goal-setting intervention, step count goals based on the past week’s daily average steps I: NR L: 16 weeks	Less	T: Fitbit tracker and Healthwatch360 app, no goal setting or peer support
Chuang 2017 Taiwan [47]	NHL	RCT	100 (50/50) (45.0)	Mean IG: 55.9 CG: 64.5	Major medical disease, MM, or bone metastasis with medical contra-indications for exercise and already practicing qigong or other exercise regular	During chemotherapy	**Unsupervised**, home-based: T: Chan-Chuang qigong program with weekly telephone calls I: NR L: 25 min, 2–3x/day D: 21 consecutive days	Less	T: UC, Nursing on side effects of chemotherapy and care
Cohen 2004 USA [48]	Lymph, HL, NHL	RCT	39 (20/19) (30.8)	Mean 51	Major psychotic illness, <18 years	During and after chemotherapy	**Supervised**, hospital-based: T: Group-based Tibetan yoga program I: NR L: 7x/week D: 7 weeks	Less	T: UC
Coleman 2003 USA [49]	MM	RCT Pilot Feasibility	24 (14/10) (41.7)	Mean 55 (42–74)	NR	During chemotherapy and ASCT	**Unsupervised**, home-based: T: Aerobic (walking, running, or cycling) and strength training (exercise bands), exercise log I: Borg Scale 12–15 L: Approx. 50 min, individual frequency D: 26 weeks	Moderate	T: UC, Encouragement to remain active and walk
Coleman 2012 USA [50]	MM	RCT	187 (95/92) (41.7)	Mean IG: 56.0 (25–76) CG: 56.4 (35–76)	Unable to understand intent of the study, major psychiatric illness, or presence of microcytic or macrocytic anaemia, uncontrolled hypertension, RBC transfusions within two weeks of study enrolment, or recombinant epoetin alfa within eight weeks of study enrolment	During unspecified intensive treatment (PBSCT)	**Unsupervised**, home-based: T: Individualized combination of stretching, aerobic exercise (walk, jog on treadmills) and strength resistance training (exercise bands), exercise log I: Aerobic:65–80% max HR, Borg Scale 11–13. Strength: 60–80% of 1 RM L: Individual length and frequency D: 15 weeks	Moderate	T: UC, Recommendation to walk L: 20 min, 3x/week
Courneya 2009 Canada [51]	Lymph, NHL indolent, NHL aggressive, HL	RCT	122 (60/62) (41)	Mean 53.2 (18–80) >60: n = 49	Uncontrolled hypertension, cardiac illness, resides >80 km from facility, not approved by oncologist	Before, during and after chemotherapy	**Supervised**, hospital-based: T: Aerobic (ergometer) I: Initial 60% of VO_2_peak, progressing 75% L: 15–45 min, 3x/week D: 12 weeks	Moderate	T: UC, Supervised exercise L: 12 sessions, 1 month, after postintervention assessments
Defor 2007 USA [52]	AA, ALL, AML, MDS, CML, NHL/HL, other malignancies	RCT	100 (51/49) (39.0)	Median 47 (18–68) IG: 46 (18–68) CG: 49 (22–64)	Unavailable treadmills at hospital admission (n = 21 excluded)	During and after allo-HSCT From transplant admission to day 100 posttransplant	**Partly supervised** Supervised, hospital-based: T: Individualized treadmill I: Comfortable speed L: 15 min, 2x/day **Unsupervised**, homebased: T: Walking I: Comfortable speed L: 30 min, 1x/day	Moderate	T: UC, Not asked to perform any formal exercise
Eckert 2022 USA [53]	BMT patients	RCT Feasibility	72 (33/39) (55.6)	NR	Engaged in yoga in past year, history of recurrent falls (>two falls in 2 months), residency outside USA, participation in a previous study with the research team, ECOG 3 questionnaire score > 3, pregnant	After ASCT	**Unsupervised**, home-based: T: Online Hatha yoga program I: NR L: Min. 60 min/week D: 12 weeks	Less	T: Online cancer health education podcasts L: 60 min/week D: 12 weeks
Furzer 2016 Australia [54]	NHL, HL, MM	RCT	37 (18/19) (NR)	Mean 48.9 (22–68) IG: 48.2 (22–64) CG: 49.6 (25–68)	Hematologist did not approve exercise due to identified risks	After chemotherapy or radiation or HSCT	**Partly supervised**, in local gyms and clinics: T: Aerobic (individual) and resistance training (machines and dumbbells) I: Cardio: 50–85% of HRmax, RPE of 10–16. Resistance: Initial 3 sets of 10–15 rep at 50% of 1 RM to 2–3 sets of 6–8 rep at 80% of 1 RM L: Max. 30 min, 3x/week D: 12 weeks	Moderate	T: Diary and general healthy lifestyle advice
Gallardo-Rodriquez 2023 Mexico [55]	ALL	RCT Pilot 3-arm	33 (11/11/11) (66.7)	Mean 23.7 (18–45) CEG: 20.5 (18–36) REG: 22.5 (18–36) CG: 28.0 (18–45)	Neutropenia, infections, bleeding at admission, were nonmotile or unable to carry out exercise; with a CNS disease preventing movement, alterations of heart function, with bone marrow or CNS relapse, with a referral from another hospital	During chemotherapy treatment	**Supervised**, hospital- and home-based: T: Cross-training or resistance (weights) exercises I: RPE of 3–6 (50–75% of HRR), 3–5 sets of 8–15 reps L: 30–50 min, 3–5x/week D: During inpatient period	Moderate	T: Mobilization I: Low L: 30 min, daily
Hacker 2017 USA [56]	ALL, AML, CLL, CML, HL, NHL, MM, MDS	RCT	67 (33/34) (38.8)	Mean 53.3 IG: 51.9 CG: 54.6	Significant comorbidity, like impending pathological fracture, making exercise potentially unsafe	During and after ASCT or allo-HSCT	**Partly supervised** Supervised, hospital-based: T: Progressive resistance training (elastic resistance bands and body weights) I: Moderate intensity, Borg scale 13 L: 2–3x/week, D: During inpatient period Unsupervised, home-based: T: Progressive resistance training (elastic resistance bands and body weights) I: Moderate intensity, Borg scale 13 L: 2–3x/week D: 6 weeks after discharge	Moderate	T: UC, Attention control with health education
Hacker 2022 USA [57]	MM	RCT Pilot	32 (17/15) (34.4)	Mean 62.78 IG: 62.21 CG: 63.44	NR	After ASCT and after discharge	**Partly supervised** Supervised, hospital-based: T: Weekly goal setting, daily step tracking, and individualized coaching I: NR L: Daily Unsupervised, home-based: T: Free-living PA, step trackers I: NR L: Daily D: 6 weeks	Moderate	T: UC, Recommendations regarding rest, PA, and exercise
Hathiramani 2020 UK [58]	Lymph	RCT	46 (23/23) (63)	Mean 61 IG: 61.5 CG: 60.4	Active disease, unstable angina or unexplained electrocardiogram, poor PS (ECOG 3 or more), pregnancy, difficulty breathing at rest, persistent cough, fever or illness, or any cognitive impairment limiting the ability to give informed consent or complete questionnaires	During and after chemotherapy	**Unsupervised**, home-based: T: Individual elements of aerobic (walking), resistance training (resistance bands, body weight), core stability and stretches I: Aerobic: Moderate intensity Resistance: ACSM guidelines with 3 sets for 8–12 rep L: 50 min, 3x/week D: 12 weeks	Moderate	T: Bed or chair-based program, mindfulness-based. CD audio guidance to relaxation techniques: mindfulness meditation, breathing exercises, guided visualization and progressive muscle relaxation. I: No advice to exercise outside of normal habits, nor asked to avoid activity L: 50 min, 3x/week
Huberty 2019 USA [59]	MPN: Polycythaemia Vera, Essential, Thrombocythemia, Myelofibrosis	RCT Pilot	62 (34/28) (93.7)	Mean 56.9 IG: 58.3 CG: 55.0	Reported performing tai chi, qi gong, or yoga for ≥60 min/week, reported engaging in ≥150 min/week of PA, utilized the study’s online yoga site: Udaya.com, (accessed on 21 August 2024) syncope in the last two months, recurrent falls: ≥2 in past two months, score of ≥15 on the PHQ-9, score of >3 on the ECOG-3, pregnant, residency outside USA	During or after chemotherapy	**Unsupervised**, home-based: T: Online homebased Hatha/Vinyasa yoga I: NR L: 5–30 min, 60 min/week D: 12 weeks	Less	T: UC, Maintain usual activity
Hung 2014 Australia [60]	Lymph, ML	RCT Pilot	37 (18/19) (46)	Mean IG: 57.5 CG: 59.9	Undergoing allo-HSCT, deemed unsuitable for study participation by physicians	After ASCT	**Unsupervised**, home-based: T: Individual telephone-delivered nutrition and exercise counselling, unsupervised aerobic (walking or cycling) and resistance (sit-to-stand or free weight) I: Recommendations based on ACSM guidelines for cancer survivors L: Various length, 3–7x/week D: 12 weeks	Moderate	T: UC
Jacobsen 2014 USA [61]	ALL, CML, CLL, MDS, MM, Lymph	RCT 4-arm	711 (180/178/178/175) (43)	Median IG E: 58 (20–76) IG SM: 58 (20–75) IG E/SM: 57 (18–75) CG: 55 (19–76) >65, n = 154 (21.6%)	Orthopaedic, neurological, or other problems that prevented safe ambulation or protocol adherence, participation in another clinical trial with QoL or functional status as a primary endpoint, planned anticancer therapies other than tyrosine kinase inhibitor or rituximab within 100 days after HSCT, planned donor lymphocyte infusion within 100 days after HSCT, planned tandem transplantation	Before, during and after allo-HSCT or ASCT	**Unsupervised**, home-based: T: Self-directed exercise program, a DVD reinforcing the program, tracking of participation in exercise and/or stress management. Exercise component: Calculation of target HR and pedometer. The stress management component also included provision of a relaxation CD I: 50–75% of estimated HRR L: 20–30 min, 3–5x/week D: 180 dayS	Moderate	T: DVD with general instruction about HSCT L: 45 min
Jarden 2009 Denmark [62]	CML, AML, ALL, AA, MDS, WM, PNH, MF	RCT	42 (21/21) (38.1)	Mean 39.2 (18–60) IG: 40.9 (18–60) CG: 37.4 (18–55) Median 40.5 IG: 45.0 CG: 38.0	Prior HSCT, recent cardiovascular, or pulmonary disease, abnormal EKG, psychiatric disorder, and motor, musculoskeletal or neurological dysfunction requiring walking aids and bony metastasis. Prior to testing: Signs of infection, anaemia, neutropenia, or thrombocytopenia, disqualified or testing postponed	During allo-HSCT	**Supervised**, hospital-based: T: Multimodal program of aerobic (ergometer), resistance (machines and weights) and active exercises, progressive relaxation, and psychoeducation I: Aerobic: Low to moderate, 50–75% HR max. RPE: 10–13. Dynamic and stretching 1–2 sets, 10–12 reps. Static: 1 set, hold for 15–30 s. Resistance: Low to moderate, 1–2 sets of 10–12 reps. Progressive relaxation: low L: 60–70 min, 3–5x/week D: 4–6 weeks	Moderate	T: UC, Conventional treatment and care, standard care for PA, PT is individualized, not providing a stationary cycle unless requested L: PT < 1½ hour/week, after allogeneic HSCT (day +1)
Jarden 2016 Denmark [63] Jarden 2021 Denmark [85]	Acute Leuk, AML de novo, AML following MDS, APL, ALL	RCT	70 (34/36) (41.4)	Mean 53.1 (19.8–73.7) IG: 51.1 (19.8–70.0) CG: 55.0 (20.3–73.7)	Severe or unstable psychological, cardio-respiratory, neurological, or musculoskeletal disease, secondary active malignancy, abnormal EKG	During chemotherapy	**Supervised**, hospital-based: T: Multimodal intervention of aerobic (ergometer), strength (weights) and relaxation exercise, nutrition support, pedometer, and health counselling I: Aerobic: 75–80% of HRmax. Dynamic resistance: Moderate to hard, 2 sets, 12 reps L: 60 min, 3x/week D: 12 weeks	Extensive	T: UC
Kim 2005 South Korea [64]	AML, ALL, SAA	RCT	35 (18/17) (51.4)	Mean IG: 32.9 CG: 34.3 (20–48)	Medicated for anxiety or depression	After allo-HSCT	**Supervised**, hospital-based: T: Individual physical exercises combined with relaxation breathing I: NR L: 30 min/daily D: 6 weeks	Less	T: UC, Routine care
Knols 2011 Switzerland [65]	AML, CLL, ALL, HL, NHL, MM, Osteo-myelofibrosis, Leuk, Amyloidosis, Testicular C.	RCT	131 64/67 (41.2)	Mean 46.7 IG: 46.6 (18–75) CG: 46.6 (20–67)	GvHD except for grade I not requiring treatment, painful joints, unstable osteolysis, chronic pain, lesions of the central or peripheral nervous system, uncontrolled cardiovascular disease, thyroid disease, or diabetes	After allo-HSCT or ASCT	**Supervised**, physiotherapy practice or fitness centre: T: Individual, physical exercises with both endurance aerobic (ergometer or walking tread mill) and resistance strength (machines and dumbbells) exercises I: Individual HR (from 50–60%, increasing to 70–80% of estimated HR max) L: 30 min, 2x/week D: 12 weeks	Moderate	T: UC
Kobayashi 2020 Japan [66]	AML, DLBCL, ALL	RCT Crossover	33 (13/20) (18.2)	Mean Wii PT/Therapist PT: 44.9 Therapist PT/Wii PT: 44.6	Grade 2 or worse CTC for Adverse Events version 4.0	During chemotherapy	**Supervised**, hospital-based: T: Individual aerobic and resistance exercises using the Wii Fit balance board I: NR L: 30 min, 5x/week D: 1 week (and then crossover)	Less	T: Individual aerobic and resistance exercises. I: Aerobic: 40–60%. Resistance: Borg 11–13 L: 30 min, 5x/week D: 1 week and then crossover
Koutoukidis 2020 UK [67]	MM, Myeloma IgG, Myeloma IgA, Myeloma Light chain, non-sec/oligo-sec.	RCT	131 (89/42) (45)	Median IG: 64 (35–86) CG: 63 (40–80)	Spinal instability. Recent spinal or other surgery for pathological fractures within 4 weeks. Abnormal EKG with unexplained clinical indication after cardiological work-up. At risk of pathological fracture based on Mirel’s score. Currently enrolled in research exercise study. Unstable angina. Musculoskeletal mobility limitations. Cognitive impairment hindering completion of questionnaire	After auto-HSCT, radiotherapy or chemotherapy	**Partly supervised** Supervised, hospital-based: T: Individual aerobic (treadmill walking, ergometer, cross-trainer or stepping) and resistance (weightlifting, body weight or resistance bands) training, exercise diaries, goal setting with physiotherapist I: Aerobic: 50–75% of predicted HR max-Resistance: 10 RM L: 1x/week D: 6 months Unsupervised, home-based: T: Individual aerobic training and resistance training, exercise diaries, goal setting with PT I: Aerobic: 50–75% of predicted HR max. Resistance: 10 RM L: Max 30 min, 2–3x/week D: 6 months	Moderate	T: UC
McCourt 2023 UK [68]	MM	RCT Pilot	50 (23/27) (38)	Mean 60.4 (37–72) IG: 59.3 (37–72) CG: 61.3 (40–72)	Declined or not suitable for auto-HSCT or too close to transplantation, restricted mobility, non-English language	Before, during and after ASCT	**Partly supervised** Supervised, hospital-based: T: Aerobic (treadmill walking or ergometer) and resistance (machines and resistance bands) exercise and behaviour change support I: Aerobic: 60–80% of HRR. Resistance: 10 RM and individually tailored to progress and/or adapt to bone disease. L: 1x/week Unsupervised, home-based: T: Aerobic (walking) and resistance (resistance bands) exercise and behaviour change support, virtual I: Aerobic: 60–80% of HRR. Resistance: 10 RM and individually tailored to progress and/or adapt to bone disease L: Aerobic exercise (Phase 1 and 3): 15–40 min, 3x/week. Resistance exercise (Phase 1 and 3): 3x/week. During phase 2 (transplant admission)	Extensive	T: UC
Mello 2003 Brazil [69]	CML, AML, SAA, NHL, MDS	RCT	18 (9/9) (55.6)	Mean IG: 27.9 (18–39) CG: 30.2 (18–44)	NR	During and after allo-HSCT	**Supervised**, hospital-based: T: Individualized exercise program with active exercise, muscle stretching and treadmill walking I: Progressing, no higher than 70% of HR max L: 40 min, 5x/week D: 6 weeks	Moderate	NR
Oechsle 2014 Germany [70]	AML, NHL, MM, Germ cell	RCT Pilot	58 (29/29) (29.2)	Mean IG: 51.7 CG: 52.9	Symptomatic cardiovascular diseases, tumor infiltration of the skeletal system with risk of pathologic fractures or compression of spinal cord, epilepsy, rheumatologic diseases, BMI < 18, BMI > 30, insufficient cognitive function, inadequate knowledge of German language for questionnaire analysis	During chemotherapy	**Supervised**, hospital-based: T: Aerobic (ergometer) and strength (body weights and resistance bands) training I: Ergometer individually adjusted Strength training: Up to 20 min at 40–60% of estimated 1 RM, sets of 16–25 repetitions L: 30–40 min, 5x/week D: Median: 21 days	Moderate	T: UC, Standard PT
Pahl 2018 Germany [71]	Leuk, AML, ALL, APL, NHL, HL, T-cell lymph, WM, MM, PMF	RCT Pilot	17 (10/7) (30)	Median 55 (47–63) IG: 47 (19–62) CG: 56 (32–63)	Unstable bone metastasis, knee or hip endoprosthesis, epilepsy, pacemaker, severe cardiovascular disease and threshold blood-count values below safety criteria, stents, or former joint injuries	During chemotherapy	**Supervised**, hospital-based: T: Whole body vibration (Galileo Sport vibration platform), including three sets of two to four different exercises (body weight) I: Borg scale 14–16 L: 20 min, 3x/week D: Median: 27 days	Moderate	T: Aerobic exercise; ergometer I: Borg scale 14–16 L: 20 min.
Pahl 2020 Germany [72]	AML, ALL, CLL, CMML, MDS, Lymph, MM, MF, Septic granulomatosis Immuno-deficiency, SAA	RCT	44 (18/26) (31.8)	Median IG: 55 (50–63) CG: 56 (32–63)	Unstable bone metastasis, endoprosthesis of knee or hip, epilepsy, pacemaker, and severe cardiovascular disease	During allo-HSCT	**Supervised**, hospital-based: T: Whole body vibration (Galileo Sport vibration platform) I: NR L: 20 min, 5x/week D: 35–44 days	Moderate	T: Mobilization and stretching L: 5x/week
Persoon 2017 Netherlands [73]	MM, (N)HL, Lymph	RCT	109 (54/55) (36.7)	Median 55 (19–67) IG: 53.5 (20–67) CG: 56 (19–67)	NR	After ASCT	**Supervised**, at local physiotherapy practices: T: Aerobic interval (cycling) and resistance (machines) training, counselling sessions (5x) I: Resistance: High intensity. Week 1–12 2 × 10 rep at 65–80% of 1 RM, week 12–18 2 × 20 rep at 35–40% of 1 RM L: 60 min, 1–2x/week D: 18 weeks	Extensive	T: UC, Not encouraged to exercise, participate in sports, PT, or rehabilitation programs
Potiaumpai 2021 USA [74]	AML, ALL, CML, MDS, MM, other Lymph	RCT	35 (19/16) (45.7)	Mean 58.8 IG: 59.3 CG: 58.2	Dementia, altered mental status, severe psychiatric conditions, pre-existing comorbid conditions that would contraindicate exercise testing, concurrent non-transplant-related chemotherapy, or radiation	Before and after allo-HSCT or ASCT	**Supervised**, hospital-based: T: Multidirectional drills and walking program I: Exertion level of moderate intensity during the multidirectional drills and a high intensity during the walking portion L: 5–30 min, 3x/weeks D: Varied	Moderate	T: UC
Safran 2022 Turkey [75]	AML, B-cell ALL, T-cell ALL, MDS, NHL, MF	RCT	43 (21/22) (51.6)	Mean IG: 38 (23–63) CG: 40.5 (24–58)	<18 years, ECOG > 3, comorbidities causing fatigue (e.g., multiple sclerosis, Parkinson’s disease, heart failure), rapid deterioration of general condition (sudden uncontrolled weight loss, confused consciousness, high CRP values), brain metastases or metastases to the femur, DVT within last 6 months, neuropathy, and rejecting NMES intervention or exercise therapy	During chemotherapy, after allo-HSCT	**Supervised**, hospital-based: T: Resistance exercise (body weights and resistance bands) combined with neuromuscular electrical stimulation I: Borg scale: Initial recommended RPE is 12–13 and is increased to about 15–16. The intensity was adjusted to a target score of 12–14 (moderate level) using the RPE scale. Intensity (~RPE 15–16) and resistance were gradually increased L: 60–90 min, 2–3x/week D: 4 weeks	Moderate	T: Resistance exercise L: 40–60 min, 2–3 days/week
Schumacher 2018 Germany [76]	MM, AML/MDS, NHL Teratoma, CML, CLL	RCT feasibility	42 (19/23) (40.5)	Median IG: 56.0 (21–65) CG: 56.5 (21–65)	Lack of compliance. Intercurrent diseases, like pulmonary and cardiac insufficiency or uncontrolled infections	During and after allo-HSCT or ASCT	**Supervised**, hospital-based: T: Exergaming on Nintendo Wii for exercising ping pong, tennis, boxing, frisbee, or aerobics and balance I: NR L: 30 min, 5x/week D: During and 30 days post HSCT	Moderate	T: PT program, eccentric and concentric movements, from supine to standing, walking, stepping or treadmill walking, stretching, strength exercise i.e., elastic bands and body weight
Shelton 2009 USA [77]	Lymph, Leuk	RCT	53 (26/27) (37.7)	Mean IG: 43.7 (22–68) CG: 48.9 (29–70)	<18 years, psychiatric disorder, significant cardiovascular disease, paraplegic or hemiplegic, unable to speak or understand English	After allo-HSCT	**Supervised**, hospital-based: T: Aerobic (treadmill and ergometer) and resistance (weights and machines) exercises I: Aerobic: 60–75% of age-predicted HR max. Strength: 1–3 sets of 10 reps L: 20–30 min aerobic, resistance individual, 3x/week D: 4 weeks	Moderate	T: multidisciplinary, inpatient, educational session incl. focus on staying active, information to exercise safely
Streckmann 2014 Germany [78]	HL, B-NHL, T-NHL, MM	RCT	56 (28/28) (25)	Mean IG: 44 (20–67) CG: 48 (19–73)	Unstable osteolysis, severe acute infections, severe cardiac and pulmonary impairments, restrictions for PA	During chemotherapy	**Supervised**, hospital-based: T: Aerobic (treadmill and ergometer), sensorimotor and strength (resistance bands) training I: Initial 60–70% HR max. At the end of session 70–80%. Sensorimotor training: Progressively increasing task difficulty. Strength training: 1 min at max force L: 60 min, 2x/week D: 36 weeks	Extensive	T: UC, Standard clinical care, incl. PT
Vallerand 2018 Canada [79]	Leuk., HL, NHL	RCT	51 (26/25) (60.8)	Mean 52.6 <60: n = 33 >60: n = 18	Chronic medical condition precluding from aerobic exercise, plan of being away from home > 2 weeks, baseline exercise levels of ≥240 min. weekly	During or after chemotherapy, radiation, HSCT	**Unsupervised**, home-based: T: Tele counselling with PA guidance with a goal of increasing aerobic exercise (walking, group fitness) levels by at least 60 min/week up to 300 min/week of moderate-vigorous aerobic exercise I: Aerobic exercise: Moderate-vigorous L: Tele-health calls: Mean: 17 min, 1x/week. Aerobic: 60–300 min/week. D: 12 weeks	Moderate	T: UC, PT guidelines, goal setting of increasing aerobic exercise levels I: Aerobic exercise: moderate-vigorous L: 60–300 min/week
Waked 2019 Egypt [80]	ALL	RCT	54 (27/27) (34)	Mean IG: 33.4 CG: 32.4	Antecedent neurological, developmental, or genetic disorder. Relapsed or secondary ALL. Received testicular, mediastinal, or craniospinal irradiation. Growth hormone insufficiency, hormone therapy. Medications that interfere with lipid metabolism. Diseases affecting cholesterol metabolism such as diabetes mellitus, thyroid dysfunction, or nephrotic syndrome	After treatment	**Supervised**, hospital-based: T: Aerobic training (ergometer) I: 60% of predictive age HR max L: 30–40 min, 3x/week D: 12 weeks	Moderate	T: UC, Normal daily activities
Wehrle 2019 Germany [81]	AML, ALL	RCT Pilot 3-arm	29 (9/10/10) (41)	Median EG: 47.7 (21.9–63.4) RG: 47.4 (41.2–62.2) CG: 50.6 (35.0–58.1)	Karnofsky score < 60, uncontrolled hypertension, cardiac illness (NYHA III-IV), instable bone metastases, lack of informed consent after screening	During chemotherapy	**Supervised**, hospital-based: T: Either aerobic (ergometer or treadmill) or resistance (body weight) training I: Endurance: 60–70% of HRmax, RPE of 12–14 Resistance: RPE 12–14 L: 30–45 min, 3x/week D: 5 weeks (median)	Moderate	T: Mobilization and stretching program, I: low intensity
Wiskemann 2011 Germany [82] Wiskemann 2014 Germany [86]	AML, ALL, CML, CLL, MDS, Sec. AML, MPS, MM, other Lymph, AA	RCT	105 (52/53) (32.4)	Mean 48.8 (18–71) IG: 47.6 (18–70) CG: 50 (20–71)	NR	Before, during and after allo-HSCT	**Partly supervised**, hospital-based and home-based: T: Aerobic (ergometer/treadmill or walking) and resistance (resistance bands) exercises I: Tailored intensity. Endurance: Borg scale: 12–14. Resistance: Borg scale: 14–16 w/8–20 rep × 2–3 sets L: 20–40 min, 2–5x/week. Endurance: 3–5x/week. Resistance: 2x/week	Moderate	T/I: Recommend moderate PA, received step counters L: Same frequency of social contact as in IG. PT up to 3x/week
Wood 2020 USA [83]	AML, MDS, ALL, CML, HL, MM, MF, AA, MCL, HLH	RCT Pilot	34 (17/17) (43)	Median 52 (28–73)	Transplant ineligibility, uncertain transplant candidacy, comorbid illness that would preclude maximal effort during exercise testing or participation in regular exercise determined by the treating physician or study exercise physiologist	Before allo-HSCT	**Unsupervised**, home-based: T: Aerobic exercise (walking, jogging, running, cycling, cross trainer or stair climbing) I: 80% HR max. From week 2 Interval, 2 min 80%, 3 min low recovery L: 30 min, 3–4x/week D: Mean: 11 weeks	Moderate	T: Fitbit Surge, no further instructions and information
Yeh 2016 Taiwan [84]	NHL	RCT	108 (54/54) (44.1)	Mean 59.8 (23–90)	Major medical disease, as uncontrolled arrhythmia, hypertension, unstable angina, severe respiratory disease, acute infection, multiple myeloma, bone metastasis, psychiatric disorders. Medical contraindications for exercise, e.g., orthopaedic problems and neurologic or musculoskeletal disturbances, or already practicing qigong or other exercise training programs	During chemotherapy	**Unsupervised**, home-based: T: Chan-Chuang qigong exercise, guidance booklet and weekly phone call I: NR L: 20–60 min, 2–3x/day (max. 5 times). D: 3 weeks	Less	T: UC

**Extensiveness** of the intervention is rated as less, moderate or extensive (see Data synthesis and analysis). AA: Aplastic anemia; ADL: Activities of Daily Living; ALL: Acute lymphoblastic leukemia; Allo-HSCT: Allogeneic hemopoietic stem cell transplantation AML: Acute myeloid leukemia; APL: Acute promyelocytic leukemia; Approx: Approximately; ASCT: Autologous stem cell transplant; BIOPHENOTYPIC LEUK: Biphenotypic leukemia; BMT: Bone marrow transplantation; BURKITT LYMPH: Burkitt lymphoma, CNS lymph, CEG: Cross-training exercise; CG: Control group; CLL: Chronic lymphoblastic leukemia; CML: Chronic myeloid leukemia; CMML: Chronic myelomonocytic leukemia; cGvHD: Chronic Graft vs. Host Disease; CNS: Central nervous system; CTC: Common Terminology Criteria DLBCL: Diffuse large B-cell lymphoma; DVD: Digital Versatile Disc; DVT: Deep vein thrombosis; ECOG: Eastern Cooperative Oncology Group Performance Scale; EKG: Electrocardiogram; GvHD: Graft vs. Host Disease; HSCT: Hematopoietic stem cell transplantation; HL: Hodgkin lymphoma; HLH: Hemophagocytic lymph histiocytosis; HR: Heart rate; HRR: Heart rate reserve; HSCT: hemopoietic stem cell transplantation; IG: Intervention group; Leuk: Leukemia; Lymph: lymphoma; MCL: Mantle cell lymphoma; MDS: myelodysplastic syndrome; MF: myelofibrosis; MM: multiple myeloma; MPS: myeloproliferative syndrome; MPN: Myeloproliferative neoplasms; MSEC: Maximal short exercise capacity; (N)HL: Non-Hodgkin’s lymphoma; NMES: Neuromuscular electrical stimulation; Non-sec: Non-secretory; NR: Not reported; NYHA: New York Heart Association Functional Classification; Oligo-sec: Oligosecretory; PA: Physical activity; PBSCT: Peripheral blood stem cell transplantation; PID: Primary immune deficiency; PMF: Primary myelofibrosis; PNH: Paroxysmal nocturnal hemoglobinuria; PS: Performance status PT: Physiotherapy; RBC: Red blood cell; RCT: Randomized controlled trial; REG: Resistance exercise group, RM: Repetition maximum; RPE: Rating of perceived exertion; SAA: Severe aplastic anemia; STEER: Strength Training to Enhance Early Recovery; UC: Usual care; UE: Upper extremity; WM: Waldenström macroglobulinemia.

**Table 2 cancers-16-02962-t002:** Summary of findings. Population: Adults with hematological malignancies. Setting: Hospital-based, home-based, or a combination, other (e.g., clinics, nature). Exercise intervention: Aerobic, resistance, or combination, or CAM. Comparison: Usual care, no intervention, placebo control. Test timepoint: Post-intervention.

Outcomes	SMD (95% CI)	Participants Completed Outcome Measures, n Studies, (n)	Quality of Evidence GRADE	Comments
Physical function * 12MWT; 2MSC; 2MWT; 6MWT; Accelerometer; KPS; SWT; TUG	0.29 (0.12–0.45)	1219 (25)	⨁◯◯◯ Very low	Downgraded, due to RoB (majority of trials), Inconsistency (moderate heterogeneity: 48.17%), and risk of Publication bias (Egger’s test *p* = 0.0516)
Aerobic capacity Aerobic Power Index; Modified Balke; Modified endurance test; Power Max, Timed Stair Climb; VO2 Max; VO2 Max Relative; VO2 Peak; VO2 Peak modified	0.53 (0.27–0.79)	853 (17)	⨁⨁◯◯ Low	Downgraded, due to Inconsistency (substantial heterogeneity: 69.21%) and risk of Publication bias (Egger’s test *p* = 0.0443)
Muscle strength GRIP; Max test; Isometric Knee Extension test; STS	0.47 (0.17–0.78)	1091 (25)	⨁⨁⨁◯ Moderate	Downgraded, due to Inconsistency (substantial/considerable heterogeneity: 82.59%)
Body composition BMI; BodPod; DEXA; SECA bioimpedance, Tanita Bioelectrical impedance	0.20 (0.03–0.37)	654 (12)	⨁⨁⨁⨁ High	No change
Physical activity GLTEQ; IPAQ; PASE	0.32 (−0.00–0.65)	358 (5)	⨁◯◯◯ Very low	Downgraded, due to RoB (majority of trials), Inconsistency (moderate heterogeneity: 56.97%), Imprecision (95% CI does not exclude 0), and risk of Publication bias (Egger’s test *p* = 0.0132)
QoL Global * CMSAS; EORTC QLQ-C30; FACT; FACT-An; FACT-BMT; FACT-Leu; GLQOL; POMS; PROMIS	0.34 (0.04–0.64)	1447 (29)	⨁⨁◯◯ Low	Downgraded, due to RoB (majority of trials) and Inconsistency (substantial/considerable heterogeneity: 87.39%)
QoL Emotional CMSAS; EORTC QLQ-C30; FACT-General, FACT-Leu; Happiness Scale; NCCN Distress Thermometer; POMS; PROMIS; SF-12; SF-36	0.33 (0.05–0.60)	1764 (28)	⨁⨁◯◯ Low	Downgraded, due to Inconsistency (substantial/considerable heterogeneity: 86.93%)
QoL Functional EORTC QLQ-C30; FACT; FACT-An; FACT-BMT; FACT-Leu; FACT-TOI	0.33 (0.09–0.57)	455 (10)	⨁⨁◯◯ Low	Downgraded, due to RoB (majority of trials), and Inconsistency (moderate heterogeneity: 37.54%)
QoL Physical CMSAS; EORTC QLQ-C30; FACT-An; FACT-BMT; FACT-Leu; FACT-TOI; PROMIS; SF-12; SF-36	0.32 (0.03–0.60)	1731 (28)	⨁⨁◯◯ Low	Downgraded, due to RoB (majority of trials) and Inconsistency (substantial/considerable heterogeneity: 87.72%)
Anxiety HADS; POMS; PROMIS; STAI	0.21 (0.13–0.55)	917 (17)	⨁◯◯◯ Very low	Downgraded, due to RoB (majority of trials), Inconsistency (Substantial/considerable heterogeneity: 84.23%) and Imprecision (95% CI does not exclude 0)
Depression CES-D; HADS; POMS; PROMIS	0.37 (0.09–0.64)	919 (17)	⨁◯◯◯ Very low	Downgraded, due to RoB (majority of trials), Inconsistency (substantial/considerable heterogeneity: 76.33%), and risk of Publication bias (Egger’s test *p* = 0.0184)
Fatigue BFI; EORTC QLQ-C30; FACT-An; FACIT-F; FACT-F; MFI; MPN-SAF; 11-point rating scale; POMS; PROMIS; SCFS	0.44 (0.16–0.71)	1860 (31)	⨁⨁◯◯ Low	Downgraded, due to RoB (majority of trials) and Inconsistency (substantial/considerable heterogeneity: 87.89%)
Pain EORTC QLQ-C30; PROMIS; SF-36	0.43 (0.13–0.73)	811 (14)	⨁⨁◯◯ Low	Downgraded, due to RoB (majority of trials) and Inconsistency (substantial/considerable heterogeneity: 77.82%)

Number (no.) of participants corresponds to the number completed with data, which is not necessarily the same as the total number included in studies; * Primary outcomes; 12MWT:12 min walking test; 6MWT: 6 min walking test; 2MSC: 2 min stair climb; 2MWT: 2 min walking distance test; AAS: Active Australia Survey; Activity MET: Activity, metabolic equivalent; BodPod: Body composition tracking system; BFI: Brief Fatigue Inventory; BMI: Body Mass Index; CAM: Complementary and Alternative Medicine (e.g., yoga, qigong, relaxation, breathing exercise); CES-D: Center for Epidemiological Studies-Depression; CMSAS: Condensed Memorial Symptom Assessment Scale; DEXA: Dual energy x-ray absorptiometry; EORTC QLQ-C30: The European Organization for Research and Treatment of Cancer Core Quality of Life Questionnaire; FACT-An: The Functional Assessment of Cancer Therapy-Anemia; FACT-BMT: The Functional Assessment of Cancer Therapy-Bone Marrow Transplantation; FACIT-F: Functional Assessment of Chronic Illness Therapy-Fatigue; FACT-F: The Functional Assessment of Cancer Therapy-Fatigue; FACT-G: The Functional Assessment of Cancer Therapy-General; FACT-General: The Functional Assessment of Cancer Therapy-General; FACT-Leu: Functional Assessment of Cancer Therapy—Leukemia; FACT-TOI: The Functional Assessment of Cancer Therapy-Trial Outcome Index; GRADE: Rate of the certainty of evidence; GRIP: Isometric hand grip; GLQOL: Graham and Longman Quality of Life Scale; HADS: Hospital Anxiety and Depression Scale; GLTEQ: Godin Leisure-Time Exercise Questionnaire; GLQOL: Graham and Longman Quality of Life Scale; IPAQ: International Physical Activity Questionnaire; KPS: Karnofsky Performance Score; Max test: Maximal strength measurement; MFI: Multidimensional Fatigue Inventory; MPN-SAF: Myeloproliferative Neoplasm Symptom Assessment Form; NCCN Distress Thermometer: National Comprehensive Cancer Network Distress Thermometer; PASE: Physical Activity Scale for the Elderly; POMS: Profile of Mood States; PROMIS: Patient Reported Outcomes Measurement Information System; QoL: Health-related Quality of Life; RoB: Risk of Bias; SCFS: Schwartz Cancer Fatigue Scale; SF-12: Short Form Survey-12 items; SF-36: Short Form Survey-36-Items; STAI: Spielberger State Anxiety Inventory; STS: Sit to Stand; SWT: Shuttle Walk Test; TUG: Timed up and go; VO_2_ Max: Volume Oxygen Maximal.

**Table 3 cancers-16-02962-t003:** Feasibility and adverse events.

		Recruitment IG and CG		Retention IG and CG	Participation	Adverse Events
					IG	IG
Author Year/Country	Sample Size Estimated, n	Eligibility Assessed, n	Included, n	Completed Post-Test, n	Adherence to Exercise (%)	AE Type, n
Accogli [36] 2022, Italy	40	193	46	42	90	No AE
Alibhai [37] 2014, Canada	40	232	38	36	28	NR
Alibhai [38] 2015, Canada	72	264	81	70	54	AE: 4 grade II musculoskeletal events
Baumann [39] 2010, Germany	60	NR	64	49	NR	NR
Baumann [40] 2011, Germany	45	NR	47	33	NR	No AE
Bayram [41] 2024, Turkey	28	39	30	26	20 (IMT)	No AE
Bird [42] 2010, UK	132	158	58	46	NR	No AE
Bryant [43] 2018, USA	30	82	18	17	80	No AE
Chang [44] 2008, Taiwan	NR	28	24	22	NR	No AE
Chen [45] 2021, China	30	46	30	29	98	NR
Chow [46] 2020, USA	41	420	41	37	75	NR
Chuang [47] 2017, Taiwan	100	105	100	96	96	No AE
Cohen [48] 2004, USA	38	NR	39	30	32	NR
Coleman [49] 2003, USA	NR	NR	24	13	NR	No AE
Coleman [50] 2012, USA	200	NR	187	166	NR	NR
Courneya [51] 2009, Canada	120	1306	122	117	92	No SAE. AE: 3 back, hip, and knee pain
Defor [52] 2007, USA	NR	122	100	85	24	NR
Eckert [53] 2022, USA	NR	326	72	43	NR	No AE
Furzer [54] 2016, Australia	NR	89	44	37	91	No SAE. AE: 2 minor exercise modifications due to pre-existing knee and back injuries
Gallardo-Rodriquez [55] 2023, Mexico	114	50	33	18	NR	No (significant) AE
Hacker [56] 2017, USA	NR	118	67	67	83	NR
Hacker [57] 2022, USA	NR	45	32	30	NR	NR
Hathiramani [58] 2020, UK	46	62	46	38	NR	No AE
Huberty [59] 2019, USA	NR	260	62	48	15	No AE
Hung [60] 2014, Australia	NR	55	37	33	NR	No AE
Jacobsen [61] 2014, USA	700	NR	711	560	NR	No AE
Jarden [62] 2009, Denmark	40	82	42	34	NR	No AE
Jarden [63] 2016, Denmark Jarden [85] 2021, Denmark	70	170	70	62	71	No SAE. AE: 8: sport-related (n = 5), cardioresp (n = 5), dizziness (n = 3), gastrointestinal (n = 3), pain/discomfort (n = 2) and bruising (n = 1)
Kim [64] 2005, S. Korea	42	NR	42	35	NR	NR
Knols [65] 2011, Switzerland	128	310	131	114	85	No AE
Kobayashi [66] 2020, Japan	32	33	33	22	67	No AE
Koutoukidis [67] 2020, UK	140	313	131	99	75	No AE
McCourt [68] 2023, UK	NR	123	50	33	NR	No SAE. AE: 1 mild episode of dizziness
Mello [69] 2003, Brazil	NR	32	18	18	NR	NR
Oechsle [70] 2014, Germany	48	NR	58	48	NR	No AE
Pahl [71] 2018, Germany	NR	121	20	11	62	No AE
Pahl [72] 2020, Germany	NR	112	71	44	59	No SAE. AE: 2 sessions stopped prematurely due to knee pain and discomfort
Persoon [73] 2017, The Netherlands	120	469	109	97	86	AE: 1 strained calf muscle
Potiaumpai [74] 2021, USA	NR	57	36	32	79	NR
Safran [75] 2022, Turkey	32	77	43	31	NR	No AE
Schumacher [76] 2018, Germany	NR	49	42	31	NR	No AE
Shelton [77] 2009, USA	164	250	61	53	75	NR
Streckmann [78] 2014, Germany	184	186	61	51	65	No AE
Vallerand [79] 2018, Canada	50	407	51	51	93	No AE
Waked [80] 2019, Egypt	54	60	54	50	NR	NR
Wehrle [81] 2019, Germany	36	39	29	22	68	No AE
Wiskemann [82] 2011, Germany Wiskemann [86] 2014, Germany	NR	141	105	80	87	NR
Wood [83] 2020, USA	60	113	34	16	NR	NR
Yeh [84] 2016, Taiwan	64	118	108	102	100	No AE
Total	NR (n = 16)	7262 NR (n = 8)	3552	2924 (82.3%)	Mean: 70% (15–100) NR (n = 21)	No AE (n = 26) AE (n = 7) SAE (n = 1) NR (n = 15)

Abbreviations: AE: adverse events, CG: Control group, IG: Intervention group, NR: Not reported, SAE: Serious adverse events.

**Table 4 cancers-16-02962-t004:** Ongoing exercise randomized trials in patients above 60 years with hematological malignancies (Clinicaltrials.gov—accessed on 13 November 2023).

Trial Identifier Design	Investigator Country	Title	Diagnosis	Sample Size, n	Age	Intervention Type and Duration	Treatment Trajectory	Primary Outcome	Study Status
NCT05642884 RCT	Smith Giri USA	Prehabilitation Feasibility Among Older Adults Undergoing Transplantation	MM	30	>60 years	Home-based prehabilitation multimodal exercise program delivered using a telehealth format 8 weeks	Before ASCT	Feasibility	Recruiting 2023-07-10 Estimated completion 2025-12-31
NCT04898790 RCT	Thuy Koll USA	Improving Cognitive Function in Older Adults Undergoing Stem Cell Transplant (PROACTIVE)	Leukemia Lymphoma MM MDS MPN	88	>60 years	Partially supervised PA in the Community Health Activities Model Program for Seniors 12 weeks	Undergoing HSCT	Change in executive function and working memory	Recruiting 2021-11-18 Estimated completion 2025-04
NCT04670029 RCT	Magali Bavaloine France	Impact of an APA Program on EFS in Patients with Diffuse Large-cell B Lymphoma Treated in 1st Line (PHARAOM)	Diffuse Large B Cell Lymphoma	186	>65 years	Partially supervised adapted physical activity with aerobic and anaerobic sessions on site and at home	During treatment	To detect an absolute difference of 15% in event-free survival between groups	Recruiting 2021-09-08 Estimated completion 2029-02
NCT04057443 RCT	Maite Antonio Spain	Nutritional and Physical Exercise Intervention in Older Patients with Malignant Hemopathies	MDS LPS MM	80	>70 years	Nutritional support according to nutritional body composition parameters (Nutritional assessment and sarcopenia evaluation). Diet counselling, oral supplemented nutrition, enteral or parenteral nutrition. Exercise program with a mixed structure, designed individually with group sessions. 24 weeks, 3 days a week	During treatment	Adherence to oncological treatment from baseline to post treatment or 6 months. Difference between dose administered and prescribed.	Unknown status Start 2019-04-11 Estimated completion 2023-06-01

ASCT: Autologous stem cell transplantation, HSCT: Hematopoietic stem cell transplantation, LPS: Lymphoproliferative Syndromes, MPN: Myeloproliferative Neoplasm, MM: multiple myeloma, MDS: Myelodysplastic Syndromes, PA: physical activity, RCT: Randomized controlled trial.

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
