# Peer review of "Limited Evidence for the Benefits of Exercise in Older Adults with Hematological Malignancies: A Systematic Review and Meta-Analysis"

_cancers, 2024, doi:10.3390/cancers16172962_

Round 1

Reviewer 1 Report

Comments and Suggestions for Authors

Authors explored the field of the benefits of exercise in older adults with hematological malignancy; as the cited guidelines by Authors indicate this concept, however, limited evidence supporting the benefits of exercise in older adults with hematological malignancy are present in literature.

As a systematic review, in this paper the original and relevant object is represented by complete review of existing literature, by exploring little evidences.

In spite of negative or few consistent evidence supporting the positive effects of exercise in older people with hematological malignancy the rigorous investigation of previous literature, could indicate the utility of further investigation in this complex field.

Regarding the methodology, Overall, High risk and some concerns were mainly due to Domain 2, deviations from the intended interventions or Domain 3 (missing outcome data), missing outcome data. Authors highlighted the need for a comprehensive investigation into the potential benefits of exercise for this population.

The conclusions are consistent with the evidence of meta-analysis approach, on the light of deeply scrutinizing of potential bias of examined studies, and the  control groups of patient.

References are appropriate and carefully checked.

Quality of tables and figures is generally good, with exception of figure 2 (risk of bias)

This research is interesting and well conducted, with rigorous methodology approach.

I have not concerns and further suggestions; almost in my opinion this paper is worthy of publishing.

Author Response

Please see the attachment, for both reviewer 1 and 2. 

Reviewer 2 Report

Comments and Suggestions for Authors

Several systematic reviews have been performed on physical exercise
of people with hematologic malignancies. In this systematic review
and meta-analysis the authors investigate the existing literature
for elder patients. However, there were no studies restricted to
the investigation of elder patients with hematologic malignancies,
and the study include all invetsigations with patients >18 years.
This review is registered in PROSPERO. The findings suggest improved
quality of life with physical exercise but the influence of age
remained inconclusive.

Concerns
1. Since the influence of age remains inconclusive the title of
the review is strongly misleading. The authors should revise
the title according to their findings.
2. Materials and Methods, 2.1 Information sources and search strategy,
it is surprising that a pre-screening of the existing literature had
not been performed before the setting of the study. Also, it is
peculiar that it took 19 months to change the search strategy from
March 2019 to October 2020, and it is not clear why it took 3yrs and
5 months, till March 2024 to perform an updated search. Why there were
so many and long in duration gaps in the investigation?
3. S1, S2, S3 and Figure 1 exhibit peculiar differences of the
initial search outcome. The S1 search returned 3468 studies, the S2
2961, the S3 1464, and finally the Figure 1 presents 3838 studies. Why?

Author Response

Please see the attachment for both reviewer 1 and 2

Round 2

Reviewer 2 Report

Comments and Suggestions for Authors

In this systematic review and meta-analysis the authors investigate
the existing literature for elder patients. The findings suggest
improved quality of life with physical exercise but the influence
of age remained inconclusive. However, the author describe the current
situation and the need for more randomized clinical trials on this
subject to conclude.